# CageNeRF: Cage-based Neural Radiance Field for Generalized 3D Deformation and Animation

**Yicong Peng**[1]    **Yichao Yan**[2*]    **Shenqi Liu**[2]    **Yuhao Cheng**[2]
**Shanyan Guan**[2]    **Bowen Pan**[3]    **Guangtao Zhai**[1*]    **Xiaokang Yang**[2]

[1] Institute of Image Communication and Network Engineering, Shanghai Jiao Tong University
[2] MoE Key Lab of Artificial Intelligence, AI Institute, Shanghai Jiao Tong University
[3] Alibaba Group

`{jack-sparrow,yichaoyan,lsqlsq,chengyuhao,shyanguan,zhaiguangtao,xkyang}@sjtu.edu.cn`
`bowen.pbw@alibaba-inc.com`

## Abstract

While implicit representations have achieved high-fidelity results in 3D rendering, it remains challenging to deforming and animating the implicit field. Existing works typically leverage data-dependent models as deformation priors, such as SMPL for human body animation. However, this dependency on category-specific priors limits them to generalize to other objects. To solve this problem, we propose a novel framework for deforming and animating the neural radiance field learned on *arbitrary* objects. The key insight is that we introduce a cage-based representation as deformation prior, which is category-agnostic. Specifically, the deformation is performed based on an enclosing polygon mesh with sparsely defined vertices called *cage* inside the rendering space, where each point is projected into a novel position based on the barycentric interpolation of the deformed cage vertices. In this way, we transform the cage into a generalized constraint, which is able to deform and animate arbitrary target objects while preserving geometry details. Based on extensive experiments, we demonstrate the effectiveness of our framework in the task of geometry editing, object animation and deformation transfer. The code and supplementary materials are available at https://pengyicong.github.io/CageNeRF/.

## 1 Introduction

Editing and manipulating 3D object is a fundamental task for generating immerse contents, which enables numerous applications in VR/AR. Although recent advances in implicit neural rendering, such as neural radiance field (NeRF) [29], are able to produce photo-realistic modeling of static objects, how to edit the 3D content remains a challenging task. In this paper, we focus on learning a general deformation field for NeRF, enabling the deformation and animation of general objects.

Editing explicit 3D representations, such as point cloud and mesh, has been well studied. Point cloud and deformable mesh [6, 3, 13] store the geometry information in 3D space, which are directly editable and are widely utilized in computer graphics for modeling dynamic objects [30]. Cage-based deformation [20, 21, 25], for example, has been used to animate polygon mesh, where per-vertex deformation can be generated by translations prescribed on a coarse cage. While mesh and point cloud representations are relatively flexible to edit and animate, it is difficult to produce geometrically accurate and visually-faithful complex objects and scenes. Additional data such as normal maps and textures are usually required to achieve high-quality rendering results.

Recent works on implicit neural rendering, such as NeRF, employ multi-layer perceptron (MLP) as a universal approximator [15, 37], allowing them to approximate a continuous volumetric scene

---

*Corresponding authors

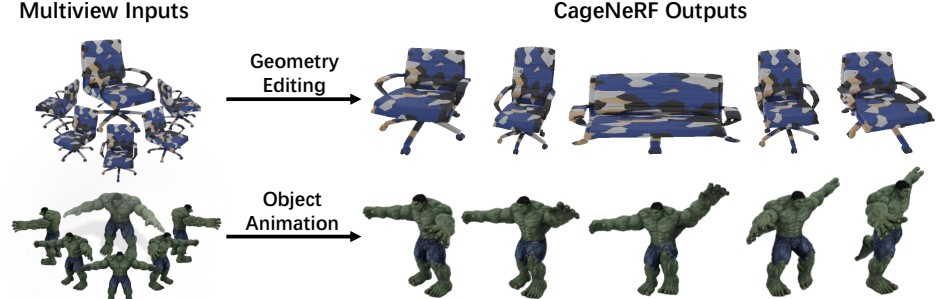

Figure 1: Given multiview images of an object, our method learns an implicit deformable 3D representation, which is used to deform and animate the object.

function to reconstruct high-fidelity results with only sparsely sampled images. Therefore, it is highly desirable if the neural rendering models can be also applied to model dynamic objects. However, optimizing a deformable radiance field end-to-end for dynamic objects like human body is a highly under-constrained problem. To address this issue, when modeling dynamic objects, most works optimize inside a deformation-invariant canonical space with data-dependent priors, such as 3DMM [4] for face deformation [1, 14], and SMPL [26] for human animation [34, 32]. While using data-dependent priors when modeling specific dynamic objects proved to be effective in providing stable correspondence between the object structure and the implicit field, it also limits the model's capability to generalize for broader applications.

To address this issue, we propose a novel framework which focuses on deforming and animating general objects represented by neural radiance fields, as shown in Figure 1. We take inspiration from the vanilla cage deforming approaches [20, 47] and extend them to work with implicit neural radiance fields. However, directly applying cage deformation into implicit field is non-trivial, due to the following issues. i) Traditional cage deformation methods work on polygon mesh with finite vertices and faces, while implicit radiance field is a continuous 3D volume. ii) Cage-based deformation transfer requires explicit geometry of the target, which is not directly accessible in the implicit field. iii) It is computationally expensive to densely sample and discretize the implicit field. To address these issues, we propose to estimate a coarse geometry of the target and generate a discrete field which supports bi-directional mapping between the canonical space and the deformed space, while keeping the radiance field continuous. Moreover, we further design a novel sampling strategy to reduce the computational cost while maintaining the render quality.

The advantages of our framework are three-fold. First, by utilizing category-agnostic cage-based deformation, our framework does not rely on data-dependent priors and can be generalized to work with a wide range of objects. Second, since cage-based deformation is a self-contained method, once the model is trained on the static scene, no additional training is required to apply deformation. Third, both cage deformation and volumetric rendering operate on the Euclidean space, we can design our framework in a decoupled manner as long as we keep the underlying coordinate system consistent between the two modules. This increases the flexibility and versatility of our framework. Moreover, as a well-developed method in computer graphics, cage-based deformation can be used in various tasks, which further expands the application scenario of our framework. We demonstrate the effectiveness of our framework in geometry editing, object animation, and deformation transfer.

## 2  Related Work

**Novel View Synthesis.** Traditional methods for novel view synthesis rely on explicit representations of 3D scenes. In these methods, 3D objects are modeled by meshes, point clouds [50, 38], etc, and then novel views are generated by shading and rendering. However, high-fidelity textures and well-designed 3D models are required for photo-realistic rendering, which makes the procedure highly expensive and inefficient. Recently, NeRF [29] has achieved impressive performance on novel view synthesis, where 3D objects are represented by implicit functions which map positions to densities and colors. A number of follow-up works have made further improvements to NeRF

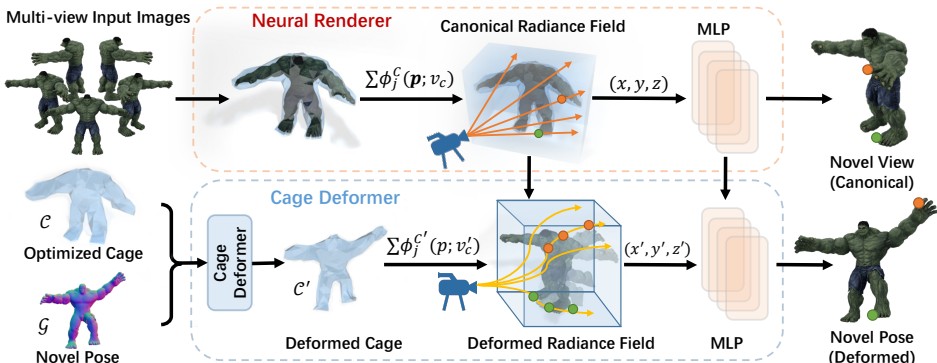

Figure 2: **Overview of CageNeRF.** CageNeRF consists of two modules: (1) a neural rendering module, which learns a radiance field from multi-view input images in the canonical space; (2) a cage deformer module that produces general cage deformation to manipulate the learned radiance field. CageNeRF deforms the implicit radiance field in an end-to-end manner.

to yield synthesis results with higher quality [53, 2, 52, 45]. Although these NeRF-based implicit methods are good at representing static scenes, it is still challenging to model dynamic scenes.

**Explicit Geometry Deformation.** Modeling deformable 3D objects is a fundamental task in computer graphics and computer vision, and triangle meshes are most commonly utilized due to their flexibility in manipulation. Traditional methods achieve this by deforming a fixed set of vertex points and then interpolating values among them [22, 46], such as skinning on skeletons [17, 28] or cages-based methods [21, 40, 56]. Along with the advances of large-scale datasets [5, 7, 51], learning-based methods are proposed by analyzing the prior principle of mesh deformation [9, 47, 10, 19]. Wang et al. [47] proposed a differentiable cage-based deformation module to synthesize shape variations and perform deformation transfer. However, these deformation methods can be only applied to explicit models, e.g., meshes. It is expensive to generate high-fidelity textures for photo-realistic rendering, and these methods cannot be directly applied to implicit models.

**Implicit Geometry Deformation.** Compared to explicit representations that only discretely control the vertex of meshes, the deformation of implicit geometry is more challenging, since the space is represented with a continuous function. Recently, various approaches build different deformation fields on implicit representations for animation [31, 18, 42, 43]. However, the generalization abilities of these methods are limited with respect to novel poses and different objects. Referring to the widely concerned digital humans, a vast range of methods are proposed for the deformation of the implicit face [12, 57, 14] and body [34, 8, 36, 33]. However, these methods heavily rely on parametric models as priors, such as 3DMM [4, 24] and SMPL [26], which are difficult to transfer to non-humanoid objects. In this paper, we propose a cage-based implicit geometry deformation method, which utilizes a sparse neural cage as guidance to build the deformation field for NeRF. The proposed CageNeRF is a general algorithm which is applicable to a wide variety of objects.

## 3 Methodology

We propose a novel framework, dubbed **CageNeRF**, for editing and animating arbitrary 3D objects represented by a neural radiance field. Radiance fields can render photo-realistic content. However, it remains challenging to perform deformation on radiance fields, since it's non-trivial to learn correspondence between implicit representations. We address this challenge by employing the cage-based deformation, which offers an elegant way of manipulating points inside a continuous space through a predefined coarse triangle mesh called cage. As shown in Figure 2, our framework consists of two modules: the neural rendering module optimizes the radiance field of a given static object inside canonical space using multiview images with known camera parameters. Then the cage deformer module deforms the cage inside this canonical space and reconstructs the implicit radiance field by translating points to novel positions.

## 3.1 Canonical Space Neural Rendering

We first train our neural renderer module to acquire the NeRF volume [29], which acts as a reference canonical space. NeRF models the scene along with its geometry, lighting, and material as a continuous volume and approximates it as a 5D radiance field using an MLP network. Given a 3D position of a point $\mathbf{p} = (x, y, z)$, the network learns a mapping function of its density $\sigma$ and color $\mathbf{c} = (r, g, b)$ variations with respect to the view direction $\mathbf{d} = (\theta, \phi)$:

$$\mathbf{F_\Theta} : (\gamma(\mathbf{p}), \gamma(\mathbf{d})) \to (\mathbf{c}, \sigma), \tag{1}$$

where $\gamma(\cdot)$ is a positional embedding function of sinusoidal basis and $\Theta$ is the trainable parameters. Then, volumetric rendering is performed to synthesize images by marching camera rays $\mathbf{r} = \mathbf{o} + t\mathbf{d}$ from viewpoint $\mathbf{o}$, with direction $\mathbf{d}$, through the radiance field. The pixel color $\hat{C}$ is estimated using discrete integration on points sampled along each ray:

$$\hat{C}(\mathbf{r}) = \sum_{i=1}^{N} \exp(-\sum_{j=1}^{i-1} \sigma_j \delta_j)(1 - \exp(-\sigma_i \delta_i))c_i, \tag{2}$$

where $\delta_i = t_{i+1} - t_i$ is the interval between adjacent sample points. We then optimize our neural render module inside this canonical space using $L_2$ loss calculated between the estimated color $\hat{C}(\mathbf{r})$ and ground truth pixel value $C(\mathbf{r})$ of the input images:

$$\mathcal{L} = \sum_{\mathbf{r} \in \mathcal{R}} \|\hat{C}(\mathbf{r}) - C(\mathbf{r})\|_2^2, \tag{3}$$

where $\mathcal{R}$ denotes the set of rays that contributes to sampled pixel.

## 3.2 Cage Optimization

To deform the neural radiance field, we need to establish a consistent correspondence between the canonical and the deformed fields. To achieve this goal, a cage, i.e., a coarse polygon mesh inside the canonical space is first optimized to be geometrically aligned and enclose the object. Then the deformation field can be interpreted as the deformation based on the structural constraint of this optimized cage. Similar to Neural Cages, we provide our network with predefined polygon meshes as the starting point of this optimization process.

The geometry of the object inside the canonical space is represented by a polygon mesh $\mathcal{G} = \{\mathbf{v}, \mathbf{f}\}$, where $\mathbf{v}$ and $\mathbf{f}$ represent the vertex and face, respectively. When an explicit surface is not available, $\mathcal{G}$ can be obtained by applying the marching cube algorithm [27] on a radiance field or SDF [45].

The template cage $\mathcal{C}$ is a polygon mesh containing $N$ vertices, which is optimized w.r.t. the mean value coordinates (MVC) of sampled vertices of $\mathcal{G}$. Since negative MVC occurs when 1) the point lies outside the cage and 2) the cage itself is self-overlapping or highly concave. By penalizing the negative values, this optimization process automatically drives the vertices of the template cage to enclose the object inside the canonical space:

$$\min_{\mathbf{v}} |\min (\phi_i(\mathbf{v}), 0)|^2, \mathbf{v} \in \mathcal{G}, i = 1, 2, \cdots, N, \tag{4}$$

where $\phi_i(\cdot)$ is the MVC weight function w.r.t. sampled vertex $\mathbf{v}$ from polygon mesh $\mathcal{G}$ and the $i$-th vertex on cage $\mathcal{C}$.

## 3.3 Cage-based Deformation

After the cage $\mathcal{C}$ is optimized w.r.t. an object, we can perform deformation based on the target novel pose/shape and the current cage shape. While it is possible to perform the cage-based deformation by explicitly editing the cage vertices, we take inspiration from Neural Cage [47] and design a cage deformer module, which generates a deformed cage based on a novel polygon mesh $\mathcal{G}_n$ given as the deformation target. This cage deformer module makes our framework applicable to tasks like deformation transfer and object reenactment. Similar to the network proposed in Neural Cages, this module takes an encoder-decoder structure with PointNet++ [35] as its backbone to automatically calculate the displacement prescribed on each vertex of the optimized cage $\mathcal{C}$, as shown in Figure 3.

We pretrain our cage deformer in two different datasets: 1) ShapeNet [7] for deformation transfer and geometry editing on general objects; 2) an extended version of Surreal dataset [11] for better generalization ability on novel pose synthesis and animation.

When performing the cage deformation, the encoder $\mathbf{E}$ extracts geometry features from the optimized cage $\mathcal{C}$ and the given deformation target $\mathcal{G}_t$, and outputs $f_c$ and $f_t$, respectively. The

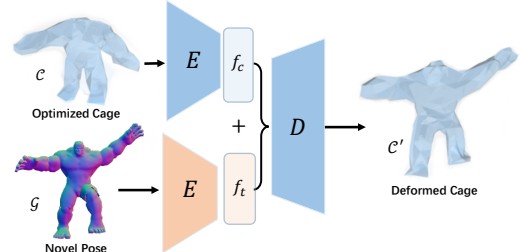

Figure 3: **Structure of the cage deformer.**

decoder $\mathbf{D}$ takes in the concatenated features and outputs the displacement $\Delta\mathbf{v}_i$ for each vertex on cage $\mathcal{C}$:

$$\Delta\mathbf{v} = \mathbf{D}(f_c \oplus f_t), \Delta\mathbf{v} \in \mathbb{R}^{N \times 3}, \tag{5}$$

where $\oplus$ refers to vector concatenation and $\Delta\mathbf{v}$ is the per-vertex displacement w.r.t cage $\mathcal{C}$. Since the cage deformation only changes vertex position while preserving its topology, the deformed cage $\mathcal{C}'$ can be obtained by adding the displacement $\Delta\mathbf{v}$:

$$\mathcal{C}' = \{\mathbf{v} + \Delta\mathbf{v}, \mathbf{f}\}, \mathbf{v}, \mathbf{f} \in \mathcal{C}. \tag{6}$$

The deformed cage $\mathcal{C}'$ is later used to generate an MVC field for radiance field editing.

### 3.4 Radiance Field Editing

Once the deformed cage $\mathcal{C}'$ is acquired, we can edit the radiance field with the following two steps. 1) Deforming the original geometry $\mathcal{G}$ acquired in Section 3.2 into a novel mesh $\mathcal{G}'$ using traditional cage-based deformation. This newly generated mesh is used as the 3D mask to filter out redundant sample points and speed up the rendering process. 2) Building a dense MVC field, which acts as the consistent correspondence during deformation, based on the deformed cage $\mathcal{C}'$ through grid sampling.

**Deformed mesh as 3D Mask.** The deformed mesh $\mathcal{G}'$ can be obtained by performing cage-based deformation directly on the extracted mesh $\mathcal{G}$ of the object inside the canonical space. For every vertex $\mathbf{v}_i \in \mathcal{G}$, we first express its position using MVC coordinates:

$$\mathbf{v}_i = \mathcal{C}(\mathbf{v}_i) = \sum_{j=1}^{N} \phi_j^{\mathcal{C}}(\mathbf{v_i})\mathbf{v}_j, \mathbf{v}_i \in \mathcal{G}, \mathbf{v}_j \in \mathcal{C}, \tag{7}$$

where $\phi_j^{\mathcal{C}}(\cdot)$ is a series of weight functions depending on the relative position between the cage vertices $\mathbf{v}_j$ and the given mesh vertex $\mathbf{v}_i$. The detailed derivation of the 3D barycentric interpolation is given in Appendix **??**. Then, the vertices of the deformed mesh $\mathcal{G}'$ can be obtained by:

$$\mathbf{v}_i' = \sum_{j=1}^{N} \phi_j^{\mathcal{C}}(\mathbf{v}_i)\mathbf{v}'_j, \mathbf{v}_i' \in \mathcal{G}', \mathbf{v}'_j \in \mathcal{C}', \tag{8}$$

where every vertex $\mathbf{v}_i'$ on $\mathcal{G}'$ has its novel position determined by the deformed cage vertices $\mathbf{v}_j' \in \mathcal{C}'$ as its new basis.

Since our radiance field is sparsely occupied by the render target, in order to increase the rendering speed while maintaining visual quality, we use the deformed mesh $\mathcal{G}'$ as a 3D mask, where only the points within the vicinity threshold $\epsilon$ of the mesh surface are sampled during volumetric rendering.

Moreover, this explicit mesh enables us to robustly establish the backward deformation, i.e., from the deformed space to canonical space, when canonical information is not present in datasets like Human 3.6M [16]. See appendix for a detailed discussion.

**Anti-aliasing MVC Sampling.** As shown in Equations 7,8, while the coordinate representation of point changed in Euclidean space, the MVC representation remains invariant. We extend this

property on meshes into 3D volumes, which yields our editable radiance field. By utilizing this invariant correspondence, we are able to effectively edit the radiance field via volumetric cage-based deformation:

$$\mathcal{F}_d = \{\mathbf{p} = (\phi_1^{\mathcal{C}}, \phi_2^{\mathcal{C}}, \cdots, \phi_N^{\mathcal{C}}) | \mathbf{p} \in \mathcal{F}\}, \tag{9}$$

where every point $\mathbf{p}$ in this deformable radiance field $\mathcal{F}_d$ has its MVC coordinates calculated under a given cage $\mathcal{C}$. $\mathcal{F}$ is the canonical radiance field learned by our neural renderer. And the deformation in this neural volume can be carried out by replacing the original cage $\mathcal{C}$ with a novel one $\mathcal{C}'$.

Since the MVC values remain invariant once the cage is fixed, to reduce computation cost, we build a pre-sampled MVC field based on the deformed mesh $\mathcal{G}'$, as is shown in Figure 4. Specifically, we apply a grid sampling strategy to the deformed space, and the MVC of the sampled grid point $\mathbf{p}_g$ is equal to the barycentric coordinates of the closest point on the deformed mesh $\mathbf{p}_n \in \mathcal{G}'$ to $\mathbf{p}_g$.

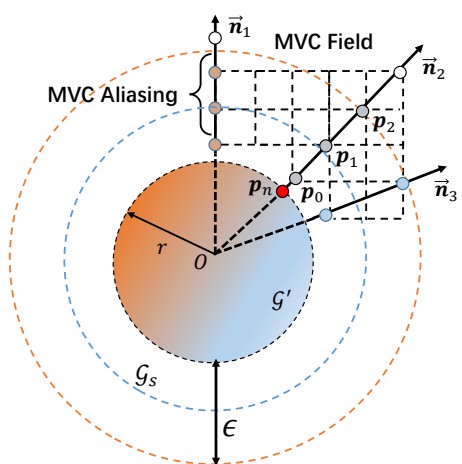

While this sampling strategy is efficient, it causes the aliasing issue as shown in Figure 4. The points $\{\mathbf{p}_0, \mathbf{p}_1, \mathbf{p}_2\}$ are sampled grid points, since they all project to the same closest point $\mathbf{p}_n$ on the deformed mesh $\mathcal{G}'$, i.e., they share the same MVC value. When rendering the deformed object, sampled points $\{\mathbf{p}_0, \mathbf{p}_1, \mathbf{p}_2\}$ are projected to the same position inside the canonical space and thus have the same color and density value. Aliased sampled points such as $\mathbf{p}_2$ have the same color and density as $\mathbf{p}_n$ instead of the correct transparent color. This causes the "expanding" effect in the deformation space, e.g., the deformed object in the figure will be rendered as a circle with radius $r + \epsilon$, which should be $r$ instead. One naive solution would be directly reducing the vicinity threshold $\epsilon$. This approach imposes a higher demand on the accuracy of both the implicit representation and the extracted polygon mesh, which is non-trivial. To mitigate this issue while maintaining a relatively large $\epsilon$, we additionally define a shell $\mathcal{G}_s$, which is also a polygon mesh that closely bounds the surface of $\mathcal{G}'$ by a small gap. This shell can be obtained by translating vertices on $\mathcal{G}'$ along their surface normal while maintaining the topology. When assigning MVC value to grid points, we compare its distance to both $\mathcal{G}_s$ and $\mathcal{G}'$ and determine the MVC value according to:

Figure 4: **Illustration of anti-aliasing MVC sample strategy in 2D case.** Grid points are projected onto the target geometry w.r.t. the surface normal $\vec{n}_i$. Given a sample range $\epsilon$, Points $\mathbf{p}_{0,1,2}$ among $\epsilon$ have the same nearest point $\mathbf{p}_n$ on deformed mesh $\mathcal{G}'$ and thus share the same MVC value. This causes aliasing when rendering the deformed radiance field. Note that Grid points outside $\epsilon$ are filtered out. Considering this problem, we devise a shell $\mathcal{G}_s$ (dotted blue circle) which closely bounds the render geometry by a small margin. Then, we assign the MVC value by comparing the distance from the grid point to both the deformed mesh $\mathcal{G}'$ and the shell $\mathcal{G}_s$ to negate this effect.

$$\phi_{grid} = \begin{cases} \phi_g, & \|\mathbf{p}_{grid} - \mathbf{p}_g\|_2 \leq \|\mathbf{p}_{grid} - \mathbf{p}_s\|_2 \\ \phi_s, & \|\mathbf{p}_{grid} - \mathbf{p}_g\|_2 > \|\mathbf{p}_{grid} - \mathbf{p}_s\|_2 \end{cases}, \tag{10}$$

where $\phi_{grid}$ is the MVC of the sampled grid point, $\phi_g$ is the MVC of the closest point $\mathbf{p}_g$ on mesh $\mathcal{G}'$, $\phi_s$ is the MVC of the closest point $\mathbf{p}_s$ on shell $\mathcal{G}_s$. When rendering the deformed neural volume, points such as $\mathbf{p}_{1,2}$ with MVC of $\phi_s$ are projected outside the object in canonical space and thus have correct transparent color.

**Deformed Rendering.** After acquiring this MVC field, we can render the deformed radiance field through volumetric accumulation. 1) Sample points along the camera ray in render space using the deformed mesh $\mathcal{G}'$ as a 3D mask. 2) Calculate the MVC value of filtered points using the MVC field via trilinear interpolation based on their coordinates. 3) Apply the cage deformation using the sampled MVC value to project points back to canonical space. 4) Infer the color and density of sampled points inside canonical space and apply volumetric rendering:

$$\mathbf{F}_\theta : (\gamma(\mathcal{C}(\mathbf{p})), \gamma(\mathbf{d})) \rightarrow (\mathbf{c}, \sigma), \tag{11}$$

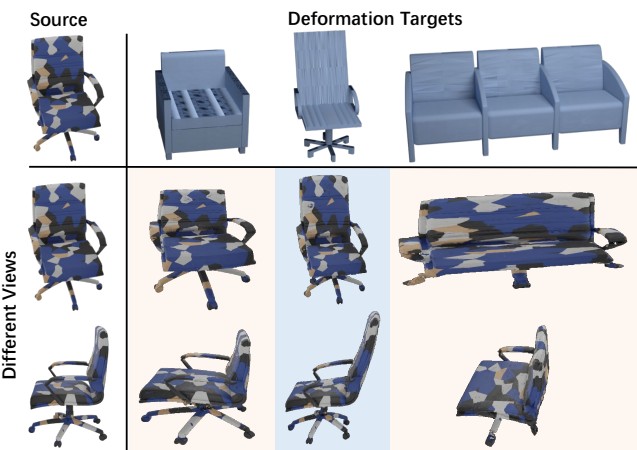

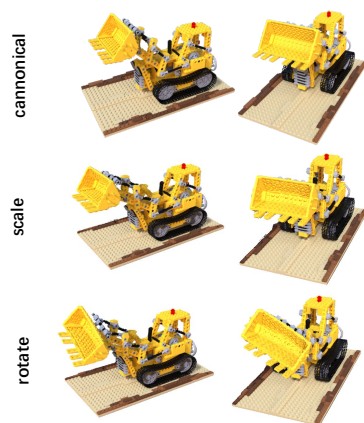

Figure 5: **Results of geometry editing via deformation transfer.** Top row shows the source object with texture and deformation target geometries. Row 2-3: deformed model under different views.

Figure 6: **Results of explicit geometry editing.** Top row shows the original object rendered in canonical space. Row 2-3: deformed model under different views.

Table 1: Animation Performance on Synthetic Datasets

| Dataset | Hulk | | | | Whale | | | | |
|---|---|---|---|---|---|---|---|---|---|
| Frame Index | 1 | 27 | 47 | Avg. | 1 | 35 | 80 | 100 | Avg. |
| PSNR ↑ | 36.434 | 35.327 | 34.078 | 35.466 | 36.146 | 35.723 | 35.755 | 35.085 | 35.742 |
| SSIM ↑ | 0.963 | 0.957 | 0.948 | 0.959 | 0.972 | 0.972 | 0.971 | 0.966 | 0.972 |
| LPIPS ↓ | 0.0126 | 0.0156 | 0.0188 | 0.0153 | 0.0117 | 0.0121 | 0.0119 | 0.0126 | 0.0120 |

where $\mathcal{C}(\cdot)$ is the cage deformation that translates sampled points back to canonical space.

## 4  Experiments

To verify the generalization of cage deformation, we evaluate CageNeRF on three tasks:

**Geometry editing (Section 4.1).** In this task, we demonstrate the capability of CageNeRF in editing the geometry of the implicit object by deforming the radiance field. This is achieved by either explicitly editing the cage shape or using the cage deformer module to generate novel cage via deformation transfer. Deformation transfer is a task that takes two objects with different geometry properties (e.g., shape, structure and scale), and modifies these properties on the source to make it resemble the target. In this task, CageNeRF is trained on the lego dataset used in NeRF [29] and a chair dataset generated in the same manner.

**Neural animation (Section 4.2).** In this task, we evaluate the continuous deformation quality of CageNeRF by comparing the results with targets animated using Linear Blend Skinning [23]. We use both synthesized objects and human videos as our benchmarks. The synthesized 3D objects contain two animated characters: 'Hulk' and a blue whale. Furthermore, we evaluate CageNeRF on real humans using Human 3.6M [16].

**Pose reenactment (Section 4.3).** In this task, we generate a synthetic dataset using a static model of a humanoid robot, which only contains polygon mesh and texture without rigged animation. This dataset contains the rendered $360°$ multi-view images with paired camera parameters. We randomly sample posing SMPL models from the Surreal dataset [44] as the reenactment target.

We further provide the ablation study w.r.t. the MVC sampling strategy. Following the majority of previous literature [29, 32, 34, 45], we use PSNR/SSIM [48] and LPIPS [54] as evaluation metrics. We refer readers to Appendix **??** for more details and results.

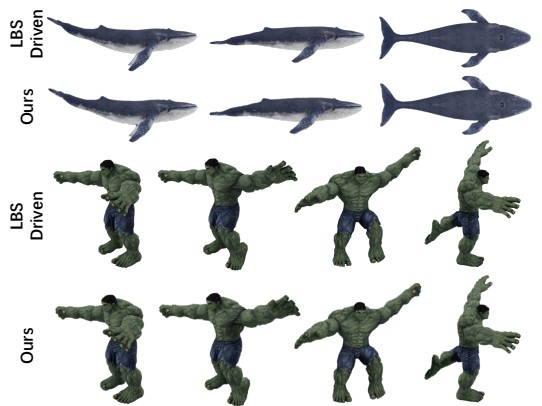
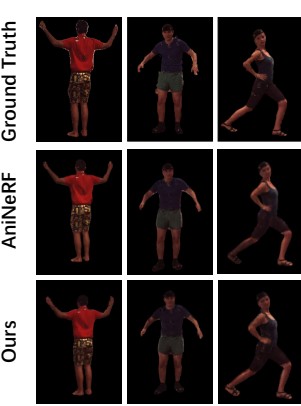

Figure 7: Qualitative results of neural animation on Whale and Hulk datasets. **Row 1, 3**: driven by linear blend skinning. **Row 2, 4**: driven by our framework.

Figure 8: Qualitative comparison of CageNeRF and AniNeRF [32] for human synthesis on Human 3.6M.

## 4.1 Geometry Editing

We demonstrate the effectiveness of deforming raidance field via explicit cage editing on the lego dataset as shown in Figure 6. We edit the cage geometry to achieve the effect of stretching and rotating the bucket. In our framework, deformation transfer can be applied to edit the geometric properties of the implicit 3D objects. Our framework is first trained on an office chair in the canonical space, as a deformation source. We randomly select objects from the ShapeNet dataset [7] as the deformation targets and then apply our cage deformer to edit the implicit geometry stored in the canonical radiance field. As shown in Figure 5, our framework is capable of editing the implicit geometry of object in the canonical space by deforming the radiance field learned by our neural renderer module. The deformed results resemble the selected target in aspects of size, shape, and scale. These results demonstrate that our framework can edit the radiance field such that the deformed geometry resembles the target.

## 4.2 Neural Animation

**Synthesized Object Animation.** To quantitatively determine the animation precision, we analyze the rendering quality of an animated sequence and compare it to the traditional linear blend skinning (LBS) animation method which operates on the explicit deformable mesh. We train our neural renderer using multiview images of the object rendered at the starting frame. To animate the object, we extract the animated mesh from the animation sequence and use it as the deformed geometry $\mathcal{G}'$. Our cage deformer then builds the MVC field using Equation 10. We animate our object by applying the extracted meshes $\mathcal{G}'$ sequentially to our framework.

As is shown in Figure 7, our framework is capable of deforming the radiance field and generating continuous animation of objects. CageNeRF also achieves nice render quality which is comparable to that driven by explicit animation methods. As shown in Table 1, our framework achieves promising results in animating the neural radiance field when compared to explicit mesh deforming methods.

**Human Animation.** We also evaluate on a real-world dataset (i.e., Human 3.6M), and compare with the state-of-the-art method AniNeRF [32] as well as other human sythesis methods [41, 49]. The Human 3.6M dataset is captured by 4 synchronized cameras and has 2D and 3D joint annotations. We first estimate the geometry proxy of the posing human using the SMPL model by optimizing the 3D joint error and 2D re-projection joint error. Then, we utilize the geometry proxy as the "novel pose" and apply the operation in Figure 3 to generate a deformed cage with the matching pose. Then we build the MVC field using this deformed cage and posing geometry proxy. Since our model only stores the information inside the canonical space, CageNeRF can generalize to novel poses without additional training. This is in contrast to AniNeRF, which needs to optimize an additional blend weight network using the estimated geometry in order to synthesize novel pose results.

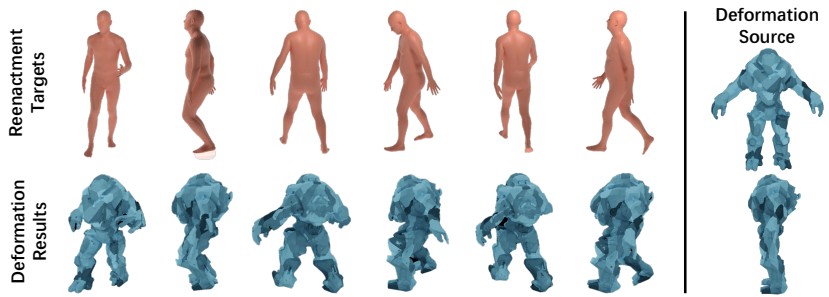

Figure 9: **Qualitative results of animation reenactment using SMPL models as the reenactment target. Top Row** shows SMPL models sampled from Surreal dataset as the reenactment target. **Bottom Row** reenactment results defromed from the souce model shows on the right.

Table 2: **Quantitative performance comparison on Human 3.6M dataset.** "NT", "NHR" and "AniNeRF" refer to Neural Textures, Neural Human Rendering and Animatable NeRF respectively.

|  | PSNR | | | | SSIM | | | |
|---|---|---|---|---|---|---|---|---|
|  | NT [41] | NHR [49] | AniNeRF [32] | Ours | NT | NHR | AniNeRF | Ours |
| S5 | 19.87 | 20.64 | 23.27 | 23.01 | 0.855 | 0.872 | 0.892 | 0.877 |
| S7 | 20.47 | 20.29 | 22.50 | 22.07 | 0.856 | 0.868 | 0.890 | 0.873 |
| S9 | 22.96 | 23.04 | 24.72 | 23.58 | 0.873 | 0.879 | 0.908 | 0.886 |
| S11 | 21.71 | 21.91 | 24.55 | 24.19 | 0.859 | 0.871 | 0.902 | 0.891 |
| Avg. | 21.25 | 21.47 | 23.76 | 23.21 | 0.861 | 0.873 | 0.898 | 0.882 |

As shown in Figure 8 and Table 2, the performance of our framework in synthesizing dynamic human body is comparable to AniNeRF across different subjects. CageNeRF achieves slightly lower PSNR due to the inaccurate estimation of the geometry used in our cage deformer module. Moreover, we employ SMPL model as the geometry proxy during cage deformation, where SMPL only models human body, while the performer in Human 3.6M dataset wears loosely fitted clothes which has complex nonrigid deformations during motion. The inaccuracy in the geometry proxy further induces sampling displacement during volumetric rendering and thus impacts the performance of our framework. Despite the fact that our framework only takes a coarse estimation of the object, it is still able to correctly deform and synthesize complex dynamic objects such as the human body.

### 4.3 Pose Reenactment

To analyze the generalization ability of CageNeRF, we apply our framework to tasks of pose reenactment and motion retargeting from humans to humanoid objects. We keep the same training process as Section 4.1. Afterward, we randomly sample posed SMPL models from the Surreal dataset as the target for deformation transfer. Both the posed SMPL model and the optimized robot cage are fed into our cage deformer to calculate the cage deformation. Finally, the barycentric interpolation is applied to deform the static robot model using Equation 8. As is shown in Figure 9, although the structure of the source robot is notably different from the target human, the deformed pose and geometric structure correspond well to the given target, while keeping the source identity and texture.

### 4.4 Impact of the MVC Sampling Strategy

To evaluate the effectiveness of our anti-aliasing MVC sampling strategy, we generate an MVC field using different sampling strategies and qualitatively and quantitatively compare the rendering results. As shown in Figure 10, the rendering result generated with grid-sampled MVC field displays inflated geometry and blurred texture, due to the inaccurate mapping between the canonical and the deformed space. In contrast, the result rendered using our anti-aliasing sampled MVC field has notably sharper textures and a more accurate geometry structure. As also shown in Table 3, the anti-aliasing sampling strategy significantly increases the render quality on all three evaluation metrics.

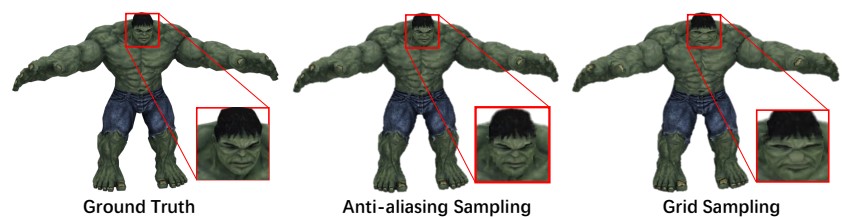


Ground Truth      Anti-aliasing Sampling      Grid Sampling


Figure 10: **Comparison between Anti-aliasing Sampling and Grid Sampling.** Our anti-aliasing sampling strategy achieves notably better rendering results w.r.t. geometry and texture.

Table 3: **Quantitative results of animated hulk using different MVC sampling strategies.** AA Sampling: the anti-aliasing sampling strategy.

| Metrics | Grid Sampling | AA Sampling |
|---------|---------------|-------------|
| PSNR | 30.695 | 35.280 |
| SSIM | 0.916 | 0.956 |
| LPIPS | 0.0293 | 0.0156 |

## 5 Limitations

Our approach achieves high quality results in deforming and animating implicit objects by combining cage-based deformation with neural radiance field. However, there remains a few limitations of our approach we would like to address. 1) While the cage-based deformation is category-agnostic and easy to generalize under various settings, it is not suited for modeling complex and subtle deformations such as facial expressions. Our work is a proof of the concept that explicit deformation can be applied to implicit neural representations. We believe our framework is also compatible with other explicit deformation schemes, which may provide the extra capability in adapting to complex movements. 2) Since the color and density of the canonical radiance field are relatively stable, our framework is not suited for modeling dynamic objects with changing color or rendering objects under varying lighting conditions. Recent works [39, 55] have developed novel techniques on factorizing basic color and density information into several decoupling elements such as surface normal, albedo, and light visibility, this further enables material editing and neural scene relighting. It would be interesting to further improve and augment our framework to have editable material, color, and lighting by incorporating novel neural rendering algorithms and structures.

## 6 Conclusion

We propose a versatile framework for deforming and animating implicit field. Comparing to previous methods, our framework excels in aspects of generalization, structural flexibility, and editing capability. This is achieved by adopting a deformable cage as the universal constraint for our radiance field editing which does not require any data-dependent prior knowledge. We demonstrate the effectiveness of our framework in the tasks of object animation, geometry editing, and deformation transfer. Experimental results show that our framework is capable of animating and deforming arbitrary objects represented by implicit field, and produces results comparable to other state-of-the-art methods.

**Acknowledgments and Disclosure of Funding**

This work is supported by NSFC (62225112, 62101325, 61831015, 62201342, U19B2035), the National Key Research and Development Program of China (2021YFE0206700), Shanghai Municipal Science and Technology Major Project (2021SHZDZX0102), and CCF-Alibaba Innovative Research Fund For Young Scholars. Authors would like to appreciate the Student Innovation Center of SJTU for the GPU support.

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
