# OpenReview forum: "CageNeRF: Cage-based Neural Radiance Field for Generalized 3D Deformation and Animation"
_NeurIPS.cc/2022/Conference — NeurIPS 2022 Accept_

### Official Review · Reviewer_gcEA · 2022-06-21

**Rating:** 6
**Confidence:** 4
**Soundness:** 2 fair
**Presentation:** 3 good
**Contribution:** 3 good

**Summary:**

CageNeRF allows for deforming a neural radiance field of arbitrary surfaces/objects, unlike prior work that was restricted to specific categories like humans or faces by design. The deformation network is trained per object and requires training data for deformations. The method design uses an invertible/bi-directional deformation parametrization based on classic graphics knowledge, namely deformation cages. It is an extension of the prior work Yifan et al. (Neural Cages for Detail-Preserving 3D Deformations; CVPR'20) to implicit fields. This extension is non-trivial due to the volumetric and non-discretized nature of the neural radiance field. The method is evaluated on deformation transfer, reenactment, and animation. All these applications apply a deformation cage to a neural radiance field, with the deformations coming from the same sequence as the radiance field in the case of synthetic reenactment and from a different sequence/scene in all other cases.

**Questions:**

* An argument why a comparison against face-specific NeRFs is not necessary would be good.
* What is the speed of the method? By what factor does the rendering slow down due to the deformations, i.e. relative to just directly using the canonical NeRF, skipping C in Eq. 15?
* I don't understand how the grid sampling (L195) interacts with the volume rendering. The points p from the rendering are first along a straight ray. How are these then transformed via the cage into the canonical radiance field? The points p won't lie on a grid but anywhere in space, so how does the grid come into play? I understand the thresholding with epsilon.

Minor questions that are relevant for a revision but not the rebuttal:
* Eq. 5 is not easy to parse. "v" is used for the vertices of the cage template, the proxy geometry, and as any individual vertex of the proxy geometry. I cannot tell which of these "v" the "max" is over. I believe it should be the vertices of the proxy geometry.
* I also don't understand how this is turned into a neural-network optimization problem. What is the loss? If this information can be found in Neural Cage, I strongly recommend to add a Background section explaining the important parts of Neural Cage that are used by this submission.
* How is the cage deformer applied in Sec. 3.2 (L137f)? Is C_t the "Optimized Cage", G_p the "Novel Pose", and C_can the "Deformed Cage"? If so, this is not at all obvious and needs to be stated explicitly.
* In Sec. 3.3, what is the loss, during pretraining and for Sec. 3.2?
* In Fig. 3, both the lower and upper branch use E_C to my understanding. It would be helpful to the reader to visualize both in the same way, e.g. use the "E_C" block also for the upper branch instead of the words "Feature Extraction".
* In Sec. 3.3, please mention that a global PointNet++ encoder is used. It was not clear to me for a while whether the deformation might be regressed per-vertex (coordinate-based MLP) instead for all vertices at once. The supplement also gives the unclear description "to predict the vertex-wise offset for the input cage base" (L85).
* In L216ff, G_d, G_p and G_s are used incorrectly, I believe. G_d should be G_n at the very least.
* Are the numbers in Table 1 from rendering into a single view or multiple views per frame index?

Typos etc.:
* L170: Not a full sentence.
* L291: Impact

**Limitations:**

Limitations are mentioned but I believe that it should be also explicitly mentioned that the cage deformer might not generalize well to target poses outside the training distribution since there is no evidence to the contrary in the submission.

General broader impact is discussed but no potential negative societal impacts are mentioned. Deformation transfer could have the typical deep fakes issue where the identity of one person is animated by motions of another.

**Strengths And Weaknesses:**

Strengths:
* Editing the neural radiance field of arbitrary surfaces/objects is an important direction for the NeRF field.
* CageNeRF is a fairly elegant parametrization and tackles e.g. aliasing issues well.
* Only human reenactment allows for a comparison against prior work, namely AniNeRF for human-specific NeRF animation. CageNeRF performs worse but close enough in my opinion, considering that it is a general method.
* The paper is written in good English.

Weaknesses:
* The method section skips over non-obvious steps in many places, see below in Questions.
* Deformation transfer is demonstrated only qualitatively and only on the ShapeNet chair category for a handful of (random) instances. In addition, this leads to deformations that seem almost just like per-axis scalings. Other categories with more interesting deformations would show that what the method does goes beyond what an anisotropic matching of the object bounding boxes could do. I don't believe the deformation transfer results are sufficient evidence that deformation transfer actually works with the method.
* The quantitative results in Table 1 are on 3-4 specific frame indices out of dozens of indices per sequence (on top of evaluating only two sequences). That leaves the door wide open for cherry-picking a few good indices. Averages (and standard deviations) across the entire sequences should be reported.
* The deformed cage is determined by a neural network and hence might not generalize well to deformed target poses that lie far away from the training data.

---

> ### Author Response · Authors · 2022-08-02
> **Response to Reviewer gcEA.**
>
> ### Question 1: The method section skips over non-obvious steps.
> Thanks for the valuable suggestion. We have rewritten the method section to make it more understandable for readers. Specifically, we have updated the notation, the symbols, and the descriptions. Please refer to our updated manuscript.
>
> ### Question 2: Insufficient evidence for deformation transfer.
> The cage deformer module, which performs the deformation transfer task, inherits the geometry deformation capability of Neural Cages since it has a similar structural design and is pretrained under the same configurations. Additionally, instead of only deforming the geometry, our framework operates on the radiance fields, which also carries out the deformation to the color and texture of the rendered target.
> - To further demonstrate the deformation effect, we conducted another experiment on the lego dataset, widely used in the field of neural rendering. Please refer to our updated manuscript for results.
> - Pose reenactment is also achieved via deformation with source and target strictly limited to humanoid objects. As is shown in the results in both the lego dataset and the pose reenactment task deformation transfer does go beyond the anisotropic matching bounding box method.
>
> ### Question 3: Average results across the entire sequence in Table 1.
> We have added the results to our updated manuscript. We evaluate each frame by rendering 50 different views of the target and calculating the average metrics as the result for each frame. We evaluate the entire animation sequence (65 frames for the hulk dataset and 120 frames for the whale dataset) and record the average metrics in our updated manuscript.
>
> ### Question 4: Why a comparison against face-specific NeRFs is not necessary?
> Our framework is more suitable for deforming objects with articulated deformation such as human posing. It is not suitable for subtle changes on the surface due to the limitation of cage deformation.
>
> ### Question 5: Speed of the method.
> Our framework achieves the average rendering speed of 2.071 seconds per image with the resolution of $800\times800$ on our Nvidia RTX3090. The rendering speed remains roughly unchanged (2.074 seconds per image) when omitting the cage deformation $\mathcal{C}(\cdot)$, since this function only involves taking one additional matrix multiplication in our implementation, which has little to no impact on the overall performance.
>
> ### Question 6: How the grid sampling interacts with volume rendering?
> During volume rendering, we sample points along each pixel ray inside the vicinity of our 3D mask. The sampled points can be in any arbitrary position inside this continuous space. To determine the actual MVC value of the sampled points, we find the nearest 8 grid points (which form an enclosing cell around the sampled points) in our MVC grid and calculate the MVC via trilinear interpolation.

---

> > ### Comment · Reviewer_gcEA · 2022-08-08
> > **Thank you for  the response**
> >
> > The revised version is indeed clearer and I appreciate the effort.
> >
> > I have two remaining request for clarification:
> >
> > (A) Why is an explicit geometry (mesh) in addition to a canonical cage required? What does that mesh do? Why is the cage insufficient, i.e. why could we not just use the canonical cage as the (coarse) explicit mesh?
> >
> > (B) There still seems to be another missing piece in my understanding of the method. Specifically, I do not get what happens for test-time rendering, after I have obtained a deformed cage. Rendering deformable implicit scene representations starts with a ray in deformed space. That ray is then deformed into the canonical space. Is that the case here? I do not see how the final rendering operation actually happens. Does the deformation of the ray from deformed space into canonical space happen using the cage? If so, in what order are what operations applied? My best guess is this:
> >
> > Pre-computation per deformation:
> > 1) Get deformed cage with the network
> > 2) Use it to deform the canonical mesh to deformed space, obtaining a deformed mesh
> > 3) construct a second mesh (shell) around the deformed mesh
> > 4) for each voxel in the MVC voxel grid (which is to be constructed), compute the MVC w.r.t. the deformed mesh and the shell, and set the voxel MVC value to whichever of the two meshes is closer to the voxel
> >
> > Actual rendering:
> > 1) Straight ray in deformed space
> > 2) Generate points on the ray
> > 3) Use epsilon filtering to throw away points not close to the deformed cage (or the deformed mesh?)
> > 4) For each remaining point, perform trilinear interpolation in the precomputed MVC voxel grid, to obtain the MVC at the continuous location of the point
> > 5) For each remaining point, use the interpolated MVC coordinates and the canonical cage as the basis to obtain the position in canonical space
> > 6) For each remaining point, get geometry and appearance at that canonical position (using the canonical position as input to the NeRF MLP)
> > 7) Volumetric accumulation of the points using standard NeRF rendering

---

> > > ### Author Response · Authors · 2022-08-09
> > > **Response to additional comments of Reviewer gcEA.**
> > >
> > > We sincerely thank you for the review and comments, which have definitely improved the quality of this paper. We hope the following responses and results can answer your questions. We enjoyed the fruitful discussions with you, thanks again for the careful review and insightful comments.
> > > ### Question A: The function and purpose of using explicit mesh.
> > > The functionalities/purposes of using an explicit mesh are three-fold:
> > > - An explicit mesh enables the cage deformer module to automatically generate deformed cages in tasks like deformation transfer and pose reenactment, i.e., the capability of Neural Cages gets transferred to the radiance field.
> > > - The explicit mesh can be deformed into a novel shape using cage-based deformation. The deformed mesh is used in test-time rendering to act as a 3D mask with a vicinity threshold of $\epsilon$. This 3D mask can effectively reduce the sample points down to 10% of the original number required in volume rendering.
> > > - **The insufficiency of cage-only deformation** An explicit mesh enables us to robustly establish the backward deformation, i.e., from the deformed space to canonical space. Cage-based deformation can express the forward deformation correctly, i.e., from the canonical space to the deformed space, but not always vice versa. In some cases, both Neural Cages and our cage deformer module will output a deformed cage with overlapping faces or collapsed vertices. Naively taking the MVC value calculated in such deformed cages will induce artifacts, since points that reside in such regions cannot be correctly projected into the canonical space. (This property can be interpreted as：CBD using a cage with a self-overlapping structure is similar to applying a matrix that is not full rank.) We provide backward deformation results on SMPL model using cage with and without self-overlapping structures via this anonymous [link](https://postimg.cc/MfjKYBsZ).
> > > Additionally, if the cage vertices is manually edited without overlapping faces and collapesd vertices, we can generate MVC field directly based on this deformed cage, i.e., without the need of an explicit mesh, since this cage deformation is inversable from deformed space to canonical space.
> > >
> > > ### Question B:
> > > The test-time rendering process is exactly the same as you've described.
> > > We sample points within the vincinity $\epsilon$ of the **deformed mesh** in Step 3 of actual rendering.

---

> > > > ### Comment · Reviewer_gcEA · 2022-08-09
> > > > **Thank you for the clarifications**
> > > >
> > > > Thank you for these clarifications. I think that incorporating them somehow into another revised version would be helpful, as I was not confident in my understanding of test-time rendering.
> > > >
> > > > Since my concerns have been addressed and, given the already revised version, I trust the authors to improve their submission with the remaining questions I and other reviewers still had, I will increase my rating to weak accept.

---

> > > > > ### Author Response · Authors · 2022-08-10
> > > > > **Response to Reviewer gcEA**
> > > > >
> > > > > Thank you for the effort and time you contributed to reviewing our manuscript. We are glad that our response addressed your concerns.

---

### Official Review · Reviewer_krXf · 2022-07-07

**Rating:** 7
**Confidence:** 4
**Soundness:** 3 good
**Presentation:** 3 good
**Contribution:** 3 good

**Summary:**

The paper proposes a way to deform Neural Radiance Fields using a cage-based paradigm. It is based on the training of a cage deformer, optimized on the poses. Then, given a new pose, the method regresses the cage on an explicit geometry and computes a dense mean-value coordinate field to compute the new visualization. The method is tested on articulated (i.e., humanoids) and non-articulated (i.e., chairs) objects.

**Questions:**

- Why do some of the results present hole artefacts?
- Why the method is not tested on more complex movements? There is any limitation to taking them into account?

**Limitations:**

The paper fairly discusses limitations and societal impact, while on the first probably a bit more elaboration on the kind of animations admitted by the method can be discussed.

**Strengths And Weaknesses:**

== Strenghts ==

- The approach seems novel and addresses a complex problem. Providing novel tools to edit Neural Radiance Fields is a hot topic with a potentially great interest for the computer graphics community
- The method seems applicable in different contexts, with good results both for rigid objects and non-rigid objects
- The method is well discussed in its elements, providing ablation and justifying the design choices.

== Weaknesses ==
- Looking at the video, there seem to be some hole artefacts, e.g., under the lifted arm of Hulk, in the pose reenactment experiment. Is it due to the stretching introduced by the paradigm, or is it because the NERF itself does not well represent some areas?
- The method still requires access to an explicit geometry, which somehow limits the arbitrary resolution power typical of neural representations. When an SDF is used, it requires a second training for the representation.
- The shown animations are not particularly complicated; probably, the use of cages requires to limit of the intersections, but I cannot understand if the method is applicable also in more complex scenarios with more articulated movements.


== minor fixes ==
- Line 174: missing a comma after "$o$ is the camera origin"
- Line 291: typo, "Impact"

== After Rebuttal ==

After the rebuttal and the discussion phase, I am prone to keep my initial rating and vote for acceptance. For the final version, I suggest including in the main manuscript the observations discussed in the reviewers-authors discussion, especially about the limitations and trade-offs of the method.

---

> ### Author Response · Authors · 2022-08-02
> **Response to Reviewer krXf**
>
> Thank you for your valuable suggestions. We have carefully studied all the comments and have reworked the whole methods part. Below we will provide point-by-point responses to all the questions and suggestions. Please also refer to our rebuttal revision for the updated manuscript.
> ### Question 1: Hole artifacts.
> There are two reasons results in the hole artifacts.
> 1. **Insufficient MVC sampling resolution.** The MVC sampling resolution is insufficient in highly concave regions such as the armpit of hulk or bumpy regions. These regions will cause the sample grid points have MVC value sampled outside the object surface which leads to a transparent color during volume rendering. We limit the sampling resolution (grid sampling voxel size) for the sake of memory consumption and computation overhead. Increasing the sampling resolution of MVC grids will mitigate the artifacts.
> 2. **Insufficient NeRF accuracy.** NeRF itself learns the geometry of the object from images and corresponding camera poses. While NeRF is capable of producing photo-realistic results it does not have a point-perfect geometry of the target, the error induced by the estimated geometry will also cause this hole artifact. However, since we design the two modules, i.e., the neural renderer and the cage deformer, in a decoupled manner, replacing the radiance field with current SDF-based neural renderer will provide a more consistent geometry, which can also help reducing this artifact.
> ### Question 2: The requirement of explicit geometry.
> 1. The requirement limits the arbitrary resolution power typical of neural representations?
> The use of explicit geometry does not limit the resolution of neural rendering. Since the sampled points has their MVC calculated via barycentric interpolation inside the closest triangle on the polygon mesh, as long as the explicit geometry has reasonably dense and evenly distributed vertices, our framework is capable of rendering upsampled images.
> 2. Do we need a second training for the repoesentation?
> We choose radiance field as the structure of our neural renderer for demonstration purpose, since the two modules, i.e., the neural renderer and cage deformer, are designed in a decoupled manner, it is feasible to replace the radiance field with SDF and perform an end-to-end training instead.
> ### Question 3: Limitation w.r.t. complex movements.
> There is a trade-off between achieving better results in complex conditions and the ability to generalize under various settings. This work is a proof of the concept that explicit deformation can be applied to implicit neural representations. We choose cage-based deformation for its simplicity in calculation. We believe our frame work is also compatible with other explicit deformation schemes, which can also be applied using a similar strategy, which may provide the extra capability in adapting to complex movements.
> Also, just as you mentioned, the deformation required to limit intersections, a typical failure case would be a deforming humanoid intoo pose with crossed arms.

---

> > ### Comment · Reviewer_krXf · 2022-08-07
> > **Rebuttal reply**
> >
> > I thank the authors for their effort to address my concerns.
> >
> > In my initial review, I raised some doubts and limitations, that in the rebuttal have been discussed in a satisfactory way - I think these considerations should also be included in the main manuscript and are useful for readers.
> >
> > I do not feel to have further points to discuss with the authors, and looking forward to other reviewers' opinions (especially in their raised concerns about the methodology, paper presentation, and also their opinion on the limitations here discussed)!

---

> > > ### Author Response · Authors · 2022-08-09
> > > **Response to additional comments of Reviewer krXf**
> > >
> > > We sincerely thank you for providing a novel perspective to our research and valuable insights through your review and comments. We will incorporate these considerations into the manuscript to our best effort. We enjoyed the process of the author-reviewer discussion with you. We are happy that our response and updated manuscript have answered your concerns. Thanks again for the careful review and insightful comments.

---

### Official Review · Reviewer_1oeX · 2022-07-14

**Rating:** 3
**Confidence:** 4
**Ethics Flag:** Yes
**Soundness:** 2 fair
**Presentation:** 1 poor
**Contribution:** 2 fair

**Summary:**

The paper extends the NeuralCages work to NeRF to enable the deformation of implicit fields. Given a NeRF, it first extracts surface points and then optimizes a template cage to enclose the points by penalizing the negative MVC values of the sampled points. It then follows NeuralCages to deform the cage. All points in the cage can be deformed accordingly with barycentric coordinates and the vertex position of the deformed cage. The author proposes a sampling trick to handle the aliasing issue (I don't understand this part).

In general, the idea of combing NeRF and NeuralCages is straightforward. However, the presentation is far from satisfactory. The experimental results are not convincing or impressive.

**Questions:**

See weaknesses.

**Limitations:**

The authors have mentioned some limitations in supplementary materials.

**Strengths And Weaknesses:**

Strengths:
The idea of introducing cage deformation to NeRF is interesting. The overall idea of combing NeRF and NeuralCages is straightforward and technically sound.

Weaknesses:
1. Although the overall idea is simple, the presentation is far from satisfactory and hard to follow. Especially, the terms and symbol system are complicated and confusing:

(a) What do the terms `geometry` and `geometry proxy` refer to? Are they refer to surface meshes only, surface points, or the regions/points within surface meshes?

(b) geometry \mathcal{G}_{can} is a set of surface points? but the geometry proxy \mathcal{G}_{p}(\mathbf{v}, \mathbf{f})  is a mesh?  What about deformed geometry \mathcal{G}_{n}? Again, the meaning of symbel \mathcal{G} is confusing.

(c) If you want to say a mesh M with vertices V and faces F, please use M={V, F}. M(V, F) seems like a function taking two arguments. For example, the text uses \mathcal{G}_{p}(\mathbf{v}, \mathbf{f}), which is strange.

(d) Line 132: which nearest neighbor algorithm? Please add a reference. Does it generate a watertight manifold mesh?

(e) Line 139: sample points from mesh surfaces or the interior. If sample points from the interior, why \mathcal{G}_{p} is closed mesh? If sample points from surfaces, can we directly sample from \mathcal{G}_{can}? And there is no need to construct \mathcal{G}_{p}?

(f) Equation 5: does \mathbf{v} refer to a sampled point? If so, please use another symbol instead of \mathbf{v}, which refers to vertices before. Also, I cannot understand why it is ``max min``? We need to penalize all negative MVC values, and it should be ``min min`` instead?

(g) Line 157: what is \mathbf{E}_{C}?

(h) For the deformed geometry \mathcal{G}_{n}, what does `n' refer to? It's kind of confusing.

(i) Line 219-220 says \mathcal{G}_{s} and \mathcal{G}_{p}. But in the caption of Figure 4, it says \mathcal{G}_{s} and \mathcal{G}_{n}. Which one is correct?

(j) Equation (14) what does `g' in \mathbf{p}_{n}^{g} refer to?

(k) Do we really need \mathcal{C}(\cdot)? Only mentioned once.

In general, section 3.2 - 3.4 is very confusing and wordy. I tried hard to follow but still failed to follow all the contents. I suggest the authors rewrite it more precisely, clearly, and compactly. A figure to illustrate the relations between the symbols may also help.

2. In section 3.3, the network architecture, loss functions, and training details are missing.

3. I can roughly understand the reason for the aliasing issue. But why do we need a vicinity threshold \epsilon? If \epsilon is very small, do we still have the aliasing issue? Moreover, I have no idea of how and why it is addressed by introducing a shell \mathcal{G}_{s}. Motivations and discussions are missing.

4. The setting of ``Synthesized Object Animation'' is kind of confusing. If we already have a sequence of animated meshes, we can directly render the images. Why do we still need the proposed CageNeRF method? I am also confused about how the method actually works (Line 256-258).

5. For Pose Reenactment, do we have meshes of humanoid objects? If so, can we directly apply NeuralCages? Why do we need CageNeRF?

6. Both Table 1 and Table 3 are computed over a single shape? What about Table 2? The experiment results are not convincing or impressive.

---

> ### Author Response · Authors · 2022-08-02
> **Response to Reviewer 1oeX**
>
> ### Question 1: The presentation.
>
> Thank you for your valuable suggestions. We have carefully studied all the comments and have reworked part of the whole method. Below we will provide point-by-point responses to all the questions and suggestions. Please also refer to our rebuttal revision for the updated manuscript.
>
> (a)
> - *Geometry* refers to the general concept of the object's shape.
> - *Geometry proxy* is a polygon mesh consisting of vertices $\mathbf{v}$ and faces $\mathbf{f}$.
> We clarify them in the revision (see Lines 125).
>
> (b)
> The symbol ${G}$ in our original manuscript refers to the geometry of the render target, which can either be a set of surface points (i.e., {G}\_{can} )  or a polygon mesh  (i.e., {G}\_{p} and {G}\_{n}). In the revision, for simplicity, we only use {G}\_{x} to represent polygon mesh.
>
> \(c\)
> We correct the notation of mesh in the rebuttal revision (see Line 125).
>
> (d)
> The used nearest neighbor algorithm is FLANN KD-tree, which can generate a watertight manifold mesh (*Muja et.al, Fast Approximate Nearest Neighbors with Automatic Algorithm Configuration.).
>
> (e)
> - We sample points from the zero-level set of the implicit SDF, i.e., the surface of the object.
> - {G}\_{p} is computed after the training of our neural renderer module. We can avoid reconstructing {G}\_{p} if we convert the sampled points from {G}\_{can} into mesh at the beginning of our pipeline.
>
> (f)
> - $\mathbf{v}$ refers to a sampled vertex instead of a sampled point.
> - maxmin and other notations are revised in Equation 5 (i.e., Equation 4 in the revision).
>
> (g)
> \mathbf{E}\_C is the pretrained encoder, which extracts features from the given mesh. For simplicity, we change \mathbf{E}\_C to \mathbf{E} Win the revision.
>
> (h)
> 'n' refers to 'novel', which means novel shape.
>
> (i)
> The caption of Figure 4 is correct. We revise the notation accordingly in our updated manuscript (see Line 210-212).
>
> (j)
> 'g' in \mathbf{p}\_{n}^{g} refer to the deformed geometry. Besides, 's' in \mathbf{p}\_{n}^{s} refers to shell. For clarity, we remove the subscript $n$ from both notations in our revision (see Line 209-212).
>
> (k)
> ${C}(\cdot)$ refers to the cage-based deformation, which is the critical component in our CageNeRF.
>
> ### Question 2: In section 3.3, the network architecture, loss functions, and training details are missing.
> Due to the page limit, the network architecture, loss function, and training details are placed into the Supp. Mat.. We will release our code upon acceptance.
>
> ### Question 3: Motivations and discussions on addressing the aliasing issue.
> 1. Why do we need a vicinity threshold $\epsilon$?
> The vicinity threshold exists to work in pair with the deformed mesh ${G}^{'}$ and serves as a 3D mask to filter out redundant sample points to speed up the neural rendering process.
> 2. If $\epsilon$ is very small, do we still have the aliasing issue?
> Decreasing this threshold will to some extent mitigate the aliasing effect. However, it will also induce other artifacts since the estimated ${G}^{'}$ does not precisely represent the actual geometry and the neural radiance field itself cannot achieve such high precision.
>
> 3. How and why it is addressed by introducing a shell {G}_{s}.
> Please refer to Section 3.4 in our revised manuscript for detailed discussion.
>
> ### Question 4: About the setting of Synthesized Object Animation.
> 1. Why do we still need the proposed CageNeRF method?
> The setting of `Synthesized Object Animation' is to offer a qualitative comparison between the linear blend skinning (skeleton driven) method and our framework under known deformation conditions. We focus on providing a novel paradigm for editing and animating the neural radiance field since most concurrent work relies on data-specific priors to perform a similar task.
>
> 2. How the method actually works (Line 256-258).
> We animate our radiance field by using the deformed mesh from each frame to deform our cage and build the deformation field accordingly.
>
> ### Question 5: For Pose Reenactment, do we have meshes of humanoid objects?
>
> 1. Do we have meshes of humanoid objects?
> Yes, we used the meshes of humanoid objects.
>
> 2. Why do we need CageNeRF?
> Neural Cage only deforms mesh. By contrast, our CageNeRF deforms the radiance field which contains both the geometry and the rendered texture.
>
> ### Question 6: About Tables 1, 2, and 3.
> - While Tables 1 and 3 are calculated on a single object, every frame has its own unique pose and shape, our method is robust under various novel poses.
> - Yes, Table 2 is calculated on a single shape from H36M dataset.
> - From Figures 5-7, we can see that CageNeRF has the ability to deform and animate the neural radiance field learned on arbitrary objects. Although there are some artifacts (e.g., subtle hole artifacts ), it should be highlighted that we provide a model to deform general objects in the radiance field, which is promising for many applications.

---

> > ### Comment · Reviewer_1oeX · 2022-08-09
> > **Thanks for the response!**
> >
> > Thanks for the response! Many errors have been corrected, and the current presentation is much clearer. After reading the response and the revised manuscript, I still have some concerns, comments, and questions.
> >
> > 1. The current version may still need another round of careful proofreading. After a quick scan of the revised manuscript, I already found many errors or inconsistencies:
> >
> > (a) Line 127, the reference of the MarchingCubes is wrong.
> >
> > (b) Line 128-134, the discussion of the cage requirement is missing. E.g., we need an optimized cage that geometrically aligns and encloses this object.
> >
> > (c) Line 202: ``expanding''
> >
> > (d) Line 255-257: "\mathcal{G}_{n}" should be "\mathcal{G}^'"?
> >
> > 2. In NeuralCage, they predict a cage for each shape using a neural network. In this work, there is no prediction, and where does the original template cage actually come from? Can we directly use \mathcal{G} as a cage? What's a good cage?
> >
> > 3. It's still unclear to me how is the shell \mathcal{G}_{s} defined. Is \mathcal{G}_{s} a sphere? Also, how does \phi_{s} is calculated? Why does introducing \mathcal {G}_{s} differ from just reducing the vicinity threshold \epsilon?
> >
> > 4. As for me, the "Synthesized Object Animation" is still not a "real application" of CageNeRF. Why do we need cage deformation here? We already have animated meshes here as the target shape. Or why do we need NeRF here? The animated meshes are already textured. Similar feeling for "Pose Reenactment", we may already have textured meshes of each pose. Do we really need to deform the NeRF?
> >
> > 5. Using one single shape for quantitative comparison is not convincing (high variance, easy to cherry-pick). I would suggest authors include large-scale experiments.
> >
> >
> >
> > Minor comments following the original review:
> >
> > 1.(d): How can we use the kNN to generate a watertight manifold mesh? The response to this point is still confusing. However, this part is replaced by MarchingCubes in the revised version.
> >
> > 1.(k): I did not mean that cage-based deformation is not a critical component. Instead, I was saying that \mathcal{C}(\cdot) should also appear in the previous definition (Equation (7)).

---

> > > ### Author Response · Authors · 2022-08-09
> > > **Response to additional comments of Reviewer 1oeX (Part 1 of 2)**
> > >
> > > Thank you for your valuable review and comments, which have improved the quality of our paper. We provide these additional responses and results which we hope can clarify your concerns. We enjoyed the effective discussion with you, despite whether this paper can be accepted.
> > > ### Question 1: Proofreading
> > > Thank you for pointing out these mistakes and inconsistencies, we will address these carefully in our manuscript.
> > > ### Question 2:
> > > - **Source of the template cage:** We use two template cages for our cage deformer module, similar to those used in Neural Cages. Specifically, we use the same 42-vertex template cage as our template for deformation transfer on objects in ShapeNet, i.e., the chair deformation. For animated hulk and the humanoid robot in pose reenactment, we use the 126-vertex template cage provided in the source code of Neural Cage.
> > > - **Prediction of canonical cage:** Canonical cage is predicted the same way as Neural Cage using the template cages mentioned above. We feed the network with the extracted mesh from canonical space as its input shape.
> > > - **Using cage versus using mesh directly:**
> > >     1. Controllable. Directly controlling a mesh with a high vertex count is computationally expensive and difficult to control. A cage is a sparsely defined mesh, which is easy to edit and control.
> > >     2. General. Cage-based deformation is a category-agnostic method that can be applied to arbitrary objects. Current mesh driving model such as SMPL requires a predefined parametric relationship between the vertices and the skeleton.
> > >
> > > ### Question 3:
> > > - **Acquisition of shell $\mathcal{G}_s$:** Shell $\mathcal{G}_s$ is generated by 1) calculate the vertex normal of $\mathcal{G}^{'}$, i.e., the geometry polygon mesh of the object in the deformed space. 2) translate every vertex along the calculated vertex normal by a small distance (0.5$\epsilon$ in our experiment). 3) construct the polygon mesh of shell $\mathcal{G}_s$ using the translated vertices and the same face definition of $\mathcal{G}^{'}$.
> > > - **The shape of $\mathcal{G}_s$:** The shape of a shell can be interpreted as an enlarged version of $\mathcal{G}^{'}$, i.e., the geometry of the object inside the canonical space. The sphere shape of $\mathcal{G}_s$ is to keep the illustration simple since the deformed mesh $\mathcal{G}^{'}$ is a sphere itself in the case of Figure 4.
> > > - **Calculating the MVC coordinates** $\phi_s$ is calculated the same way as $\phi_g$ since both of them are polygon meshes. Specifically we calculate the MVC coordinates of a grid point w.r.t. a polygon mesh inside our MVC field using the following steps: 1. find the nearest triangle on the polygon mesh to this grid point. 2. calculate the MVC coordinates of the triangle vertices. 3. using barycentric interpolation of these three vertices to determine the MVC of the grid point. In our implementation, we use functions provided by mesh processing libraries called [Mesh](https://github.com/MPI-IS/mesh) to perform barycentric interpolation and [PyMesh2](https://pypi.org/project/pymesh2/) to calculate MVC coordinates.
> > > - **Difference between reducing $\epsilon$ and applying shell:** Directly reducing $\epsilon$ will require the NeRF network to learn a more accurate representation and requires a more precise polygon mesh. Both requirements are difficult to meet while maintaining a relatively larger $\epsilon$ can achieve on-par performance without imposing higher demands on the network.
> > >
> > > ### Question 4: The Setting of Experiments
> > > The "Neural Animation" experiments are not intended to show an application when there exist both mesh and texture, but to demonstrate that our method can generate comparable animation results as LBS driven methods.
> > > The humanoid robot used in experiment "Pose Reenactment" only has a polygon mesh and is not animated or rigged. And as demonstrated in Figure 9, our deformed result can correctly take the same pose as the target.

---

> > > ### Author Response · Authors · 2022-08-09
> > > **Response to additional comments of Reviewer 1oeX (Part 2 of 2)**
> > >
> > >
> > > ### Question 5: Experiment Design and Datasets
> > > Deformable radiance field is a pioneering research topic. There lacks standard benchmarks for us to conduct experiments.
> > > As for the choosing of our datasets and experiments:
> > > - The experiment "Geometry Editing" and "Pose Reenactment" are representative in the  aspects of learning from NeRF-like datasets, and deforming both general objects and humanoids.
> > > - The experiment "Neural Animation" is conducted to demonstrate the quality and accuracy of our deformable radiance field.
> > > - We conduct our experiment on H36M dataset to demonstrate that our framework is capable of learning the canonical radiance field from the deformed space, i.e., multi-view images of posing human and deforming real-world targets.
> > > Our framework is also generalizable to standard reconstruction datasets such as DTU.
> > >
> > > ### Algorithm used in extracting mesh:
> > > The KNN method mentioned in our original manuscript is an early practice of extracting mesh implemented using the built-in function of the mesh processing library [Open3D](http://www.open3d.org/). The KNN method which yields a similar result as the marching cube algorithm. To keep simplicity, we choose the marching cube algorithm in both our manuscript and the actual experiments.
> > >
> > > ### Notation of $\mathcal{C}(\cdot)$
> > > Thank you for your suggestion, we will follow your advice and update our equations accordingly.

---

### Author Response · Authors · 2022-08-07
**Author-Reviewer Discussion**

Dear reviewers:
The author-reviewer discussion period will end in a few days, we would appreciate it if you could spare your valuable time and have a brief discussion with us.

---

### Meta-Review · Area_Chair_dnA4 · 2022-08-27

**Recommendation:** Accept
**Confidence:** Less certain

**Metareview:**

All reviewers consider the central idea of the paper to be novel and interesting, and that deformable NeRFs are a valuable research area.

All reviewers agree that the initial paper was poorly presented, and that the revised version is considerably improved.

While accepting the authors' rebuttal regarding the lack of suitable datasets to evaluate deformable NeRFs. the qualitative presentation could be improved further, and in particular, it is recommended to include video sequences of continuous deformation, which will more clearly show the artifacts of both the proposed and other methods.


**Award:**

No

---

### Decision · Program_Chairs · 2022-09-14

Accept